# Population Status of the Tropical Freshwater Shrimp *Xiphocaris elongata* in Urban and Forest Streams in Puerto Rico

Wesley X. Torres-Perez * and Omar Perez-Reyes *

Department of Environmental Sciences, University of Puerto Rico, Rio Piedras Campus, San Juan, PR 00956, USA
* Correspondence: wesley.torres@upr.edu (W.X.T.-P.); omar.perez15@upr.edu (O.P.-R.)

**Abstract:** Most of the human population lives in cities, and understanding their impact on freshwater environments is essential. Streams in cities face many environmental challenges that have been described in the concept of Urban Stream Syndrome. This concept illustrates urban streams' biological, hydrological, chemical, and physical stressors. In tropical streams, these stressors impact shrimp, fish, insects, and other macroinvertebrates that inhabit the freshwater ecosystems. Freshwater shrimp are vulnerable to urban activities, physical, chemical, and ecological impacts. For this reason, these organisms have been used as biological indicators of stream health in the tropics. The shredder shrimp *Xiphocaris elongata* plays a fundamental role in the organic matter process and decomposition. The objectives of this study were to characterize the population of *X. elongata* and to identify differences in the abundance of *X. elongata* between urban and forest streams. Our results showed that highly urbanized areas have a significantly lower abundance of the shredder shrimp *X. elongata* than medium or low urban reach in the urban and forested watersheds. This study also showed that physicochemical and geomorphological variables are important environmental factors that influence the abundance of *X. elongata* in Puerto Rican streams.

**Keywords:** decapods; shredder shrimp; tropical stream; urban watershed; Xiphocarididae





## 1. Introduction

One of the most critical socio-ecological concerns of the 21st century is the rise of urban areas as the dominant geographical context on Earth [1]. Urban activities can have significant and sometimes irreversible effects on the natural world. For example, many urban streams face environmental challenges in biological, hydrological, chemical, and physical stressors described by the Urban Stream Syndrome [2]. The active use of land stretches by humans commonly involves changes to the land-cover that cause fragmentation of the environment, changing every inherent process that supports it. According to the Urban Stream Syndrome, one of the primary drivers of degraded conditions in urban streams is the impervious surface cover that inhibits water infiltration into the soil, increasing runoff directly into drainages and streams [3]. Increased runoff in streams causes flashy hydrology with an increased frequency and magnitude of flooding [4]. This response can lead to stream bank erosion and sedimentation of stream channels [3].

In addition to changes in hydrology, urban streams can become contaminated with different chemicals, including pesticides, pharmaceuticals, personal care products, and industrial residues [4,5]. These contaminants can come from point sources, including urban and industrial wastewater effluent, and non-point sources, such as stormwater and lawn care products [6]. Several studies indicate that due to this contamination, organisms and organs can be affected detrimentally, such as their resilience to the natural disturbance regime. Natural tropical streams tend to have high densities of aquatic organisms, including freshwater shrimp, fish, and other macroinvertebrates affected by urban activities. However, there is a need to document these effects. This is particularly needed for freshwater shrimp, which are vulnerable to urban activities physical, chemical, and ecological stressors [7].

For this reason, freshwater shrimp have been used as biological indicators of stream health, and much is known about their ecology in tropical streams [8–10]. In Puerto Rico, a total of 17 species of freshwater shrimp from three families (Atyidae, Palaemonidae, and Xiphocarididae) have been described [11]. All freshwater shrimp species on the island show an amphidromous life cycle where adults reproduce and release the larvae in high elevations of the watershed. The larvae then spend time in the estuary until they metamorphose and migrate upstream [12–14].

Among the species described for Puerto Rico, the shredder *Xiphocaris elongata* [15] plays a fundamental role in organic matter decomposition. Contrary to most of the temperate streams, the functional feeding aquatic groups along Puerto Rican streams are dominated by decapods that adjust to the natural hydrological gradient of the stream based on their primary feeding behavior (e.g., predation, shredding, filtering, and grazing) and resource availability in their environment. Thus, the shredder shrimp has an essential role as an organism that connects all functional groups of food webs [16–18]. The aims of this study were: (1) to determine the population status of *X. elongata* and (2) to identify differences in the abundance of *X. elongata* between urban and forest streams. We hypothesized that there would be a decreased abundance of *X. elongata* in watersheds with different urban gradients due to alterations in their habitat. Therefore, we compared the abundance of *X. elongata* between urban (Río Piedras), and forested (Río Sabana) watersheds and between different urban impacted reaches (low urban—LU, medium urban—MU, and high urban—HU). We also compared the abundance of *X. elongata* between different rainfall seasons [low rainfall—(LRF) and high rainfall—(HRF)]. We expected a lower abundance of freshwater shrimp in watersheds with a higher and medium urban land cover because of changes in physicochemical variables.

## 2. Materials and Methods

### 2.1. Study Sites

The Río Piedras and Río Sabana watersheds were selected to describe and compare the population status of the freshwater shrimp *Xiphocaris elongata*. These watersheds represent the low (forested) and high (urban) urbanization levels based on the percentage of urban cover, respectively. In each watershed, three sampling locations were selected from the river mouth to the headwaters in different urban intensities. At each location, two 10 m sampling reaches were randomly selected (to maximize the range of habitat types: pools and riffles) to sample freshwater shrimp and measure their physicochemical parameters.

The sampling locations within the watersheds were classified as low (LU), medium (MU), and high urban (HU) reaches. High urban (HU) reaches were defined as an area adjacent to densely settled census blocks with a population of 2500 to 50,000 [19]. Medium urban (MU) or rural reaches consisted of census blocks with a population between 2500 to 1000 [19]. Finally, low urban (LU) or natural reaches were defined as areas that appeared to be minimally altered by human actions and had less than 1000 inhabitants (Table 1).

**Table 1.** Classification of the intensity of urban population in study sites reaches along the urban (Río Piedras) and forested (Río Sabana) watersheds. These sampling sites were classified under the following criteria: high urban (HU)—a total population between 2500 to 50,000; medium urban (MU)—a population between 2500 to 1000, and low urban (LU) less than 1000 inhabitants [19].

| Watershed | Total Urban Population along Reach Areas (Inhabitants) | | |
|---|---|---|---|
| | LU | MU | HU |
| Río Sabana | 877 | 1821 | 5327 |
| Río Piedras | 985 | 2515 | 30,118 |

The Río Piedras watershed (Figure 1A) has the island's highest level of urban development. It has a population of 2–2.5 million, making it the second-largest city in the Caribbean [20,21]. This watershed drains most of the San Juan Metropolitan area [22,23]. The main water source of the Río Piedras is located at about 70 m above sea level, where

the flow is regulated by Lago Las Curías [7]. This river flows into Caño Martin Peña, where it enters into the San Juan Bay Estuary, but before it passes through different urban areas of San Juan [7,20]. This watershed's land cover is comprised 55.4% urban areas, 15.3% pastures, and 26.4% forest [24]. Native trees and ornamental exotic species such as *Delonix regia*, *Tabebuia heterophylla*, *Cecropia peltata*, and *Spathodea campanulata* characterize the riparian vegetation in this stream. Freshwater shrimp were collected in the following stations: LU (18°2036.9″ N, 66°0323.3″ W), MU (18°2325.4″ N, 66°0331.6″ W), and HU (18°2410.3″ N, 66°0402.5″ W).

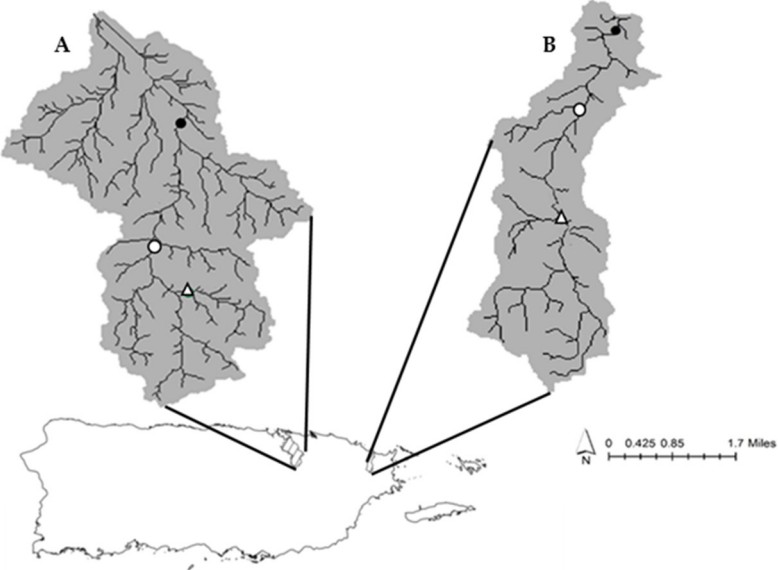

**Figure 1.** (**A**) Urban watershed (Río Piedras), (**B**) forested watershed (Río Sabana). The white triangle represents low urban reach (LU), the white circle represents medium urban reach (MU), and the black circle represents high urban reach (HU).

The Río Sabana watershed (Figure 1B) is located in the northeastern area of Puerto Rico. The upper headwaters of the watershed are managed by the U.S. Forest Service to sustain the riparian forest cover [24]. However, humans have impacted the middle and lower parts [25]. The land-cover is comprised of 71.4% forest, 15.8% pasture, and 8.8% urban areas [24]. The transition zones between forested and urban have moderate densities of native and introduced trees in contrast with the urban sections, where the valleys have high densities of non-native grasses and fruit trees. Non-native species are scarce in the headwater sites but increase downstream in the urban reaches. The urban reaches are characterized by riparian zones that are primarily composed of grasses, where the most common species are *Panicum aquaticum* and *Bambusa vulgaris*. Freshwater shrimp were collected in the following stations: LU (18°1932.7″ N, 65°4346.9″ W), MU (18°2047.9″ N, 65°4337.0″ W), and HU (18°2203.1″ N, 65°4300.9″ W).

### 2.2. Physicochemical Variables

In each watershed, stream habitats were quantified through direct measurements of the physical characteristics of the pools (length, width, depth, substrate composition, and pool canopy cover) at six reaches (two per sampling location) with low, medium, and high urban reaches areas. There were five physicochemical variables that were measured (temperature, pH, conductivity, dissolved oxygen, and turbidity) using a Hydrolab DS5X multiparameter sonde that was previously calibrated following the manufacturer's instructions.

The size composition of the substrate was estimated at each sampling reach where the decapods were collected. A total of 100 rocks from four samples of 0.5 m$^2$ were selected randomly to characterize the substrate composition using a gravelometer. The categories were expressed as a percent of bedrock, boulders, cobble, coarse gravel, fine gravel, and

sand/fines. The percentage categories were converted to a substrate index (SI) using the following formula: SI = [(0.08)(% bedrock) + (0.07)(% boulder) + (0.06)(% coble) + (0.05)(% gravel) + (0.04)(% fine gravel) + (0.03)(% sand and fines)].

The canopy coverage percentage was determined using a spherical densiometer directly in the middle of the stream.

### 2.3. Freshwater Shrimp Sampling

Freshwater shrimp were collected using an electrofishing backpack (Model 12-B, Smith-Root, Vancouver, Washington, DC, USA) [26]. Collections consisted of five upstream electrofishing passes in each sampling reach (10 m) that was bounded by a net to limit the emigration or immigration of decapods and fish during sampling. Hand nets were used to collect the organisms. The habitats that were sampled included riffles, runs, pools, and aquatic vegetation. Taxonomic identification was followed using Pérez-Reyes et al. [11]. The cephalothorax length (CL) of each shrimp was measured from the post-orbital region to the end of the carapace with a dial caliper (0.01 mm precision). The measurement from the tip of the rostrum was not used because the length of *Xiphocaris elongata* varies depending on the presence of fish predators [10,27]. Catch per unit effort (CPUE) was calculated for one-year sampling (October 2018–September 2019). Catch per unit effort represents the quantity of shrimp divided by the sampling effort (e.g., the number of passes in each sampling location).

### 2.4. Statistical Analyses

The environmental and physicochemical variables were compared among study sites for both watersheds using one-way ANOVA analysis. Two-way ANOVAs were used to compare the catch per unit effort among streams, reaches, and between two rainfall periods: low rainfall season (LRF)—December to March and high rainfall season (HRF)—April to November. Non-metric multidimensional scaling (NMDS) analysis was used to describe the relationship between the environmental variables (depth- pool, substrate index, %pool cover, dissolved oxygen, conductivity, temperature, and turbidity) and *X. elongata* abundance. Stream habitat variables were measured at the different sampling sites in (a) urban watershed (Río Piedras) (PRH—Piedras River high urban reach, PRM—Piedras River medium urban reach, and PRL—Piedras River low urban reach) and (b) forested watershed (Rio Sabana)(SRH—Sabana River high urban reach, SRM– Sabana River medium urban reach, and SRL—Sabana River low urban reach) were plotted as vectors. The Bray–Curtis dissimilarity index was used as the distance variable for NMDS ordination [28].

### 3. Results

#### 3.1. Physicochemical Variables

In the Río Sabana watershed, one-way ANOVAs for physicochemical parameters showed significant differences in temperature, pH, turbidity, and conductivity among sampling stations (Table 2). Our results showed an increased stream temperature from the headwaters to the river mouth. The lowest mean temperature was recorded in LU reach (21.3 ± 0.03 °C), while MU and HU reach had mean values of 21.7 ± 0.06 °C and 22.3 ± 0.03 °C, respectively (Table 2). In addition, significant differences in pH were observed among the reaches (Table 2). The LU reach had a more near to neutral pH mean value (7.43 ± 0.03), while no major differences were observed in the HU reach (7.58 ± 0.03). The Río Sabana watershed also had significant differences in turbidity and conductivity among the study reaches (Table 2).

**Table 2.** The mean (±SE) physicochemical measurements in low urban (LU), medium urban (MU), and high urban (HU) reaches in the urban (Río Piedras) and forested (Río Sabana) watersheds. One-way ANOVA for comparison among reaches in each stream. NS—not significant at $p = 0.05$; *** $p < 0.001$.

| Watershed | Physicochemical Variables | LU | MU | HU | ANOVA |
|---|---|---|---|---|---|
| **Urban** | Temperature (°C) | 22.4 ± 0.04 | 23.8 ± 0.03 | 23.9 ± 0.03 | *** |
| | pH | 7.9 ± 0.03 | 7.9 ± 0.02 | 8.0 ± 0.03 | *** |
| | Turbidity (NTU) | 0.2 ± 0.001 | 0.2 ± 0.001 | 0.2 ± 0.001 | *** |
| | Conductivity (µS·cm$^{-1}$) | 330 ± 1 | 337 ± 2 | 355 ± 1 | *** |
| | Dissolved oxygen (mg·L$^{-1}$) | 7.9 ± 0.1 | 7.8 ± 0.1 | 7.7 ± 0.1 | *** |
| **Forested** | Temperature (°C) | 21.3 ± 0.03 | 21.7 ± 0.06 | 22.3 ± 0.03 | *** |
| | pH | 7.4 ± 0.03 | 7.6 ± 0.03 | 7.6 ± 0.03 | *** |
| | Turbidity (NTU) | 0.1 ± 0.0004 | 0.1 ± 0.0004 | 0.1 ± 0.001 | *** |
| | Conductivity (µS·cm$^{-1}$) | 124 ± 1 | 154 ± 1 | 155 ± 1 | *** |
| | Dissolved oxygen (mg·L$^{-1}$) | 8.9 ± 0.1 | 9.0 ± 0.1 | 0.1 ± 0.0004 | NS |

One-way ANOVAs for physicochemical parameters in the Río Piedras watershed showed significant differences in temperature, pH, turbidity, conductivity, and dissolved oxygen (Table 2). The lowest significantly different mean temperature value was recorded in the low urban reach (22.4 ± 0.04 °C), while medium and high urban sites had mean values of 23.8 ± 0.03 °C and 23.9 ± 0.03 °C, respectively (Table 2). Similarly, the lowest conductivity values were also found in low urban sites (330.0 ± 1 µS·cm$^{-1}$), while the highest values were observed in high urban areas (354.0 ± 1 µS·cm$^{-1}$). Significant differences in pH values were also observed among study sites (Table 3). The low urban sites had the lowest pH mean value (7.7 ± 0.03), while the highest values were observed in high urban sites (7.9 ± 0.03).

**Table 3.** Mean (±SE) physicochemical measurements in low urban (LU), medium urban (MU), and high urban (HU) reaches in the Urb (urban stream, Río Piedras) and For (forested stream, Río Sabana) watersheds (*** $p < 0.001$).

| | LU | MU | HU | ANOVA |
|---|---|---|---|---|
| | Urb <br> For | Urb <br> For | Urb <br> For | |
| **Temperature (°C)** | 22.4 ± 0.04 <br> 21.3 ± 0.03 | 23.8 ± 0.03 <br> 21.7 ± 0.06 | 23.9 ± 0.03 <br> 22.3 ± 0.03 | *** |
| **pH** | 7.8 ± 0.03 <br> 7.4 ± 0.03 | 7.9 ± 0.02 <br> 7.6 ± 0.03 | 7.9 ± 0.03 <br> 7.6 ± 0.03 | *** |
| **Turbidity (NTU)** | 0.2 ± 0.001 <br> 0.1 ± 0.0004 | 0.2 ± 0.001 <br> 0.1 ± 0.0004 | 0.2 ± 0.001 <br> 0.1 ± 0.001 | *** |
| **Conductivity (µS·cm$^{-1}$)** | 330 ± 1 <br> 124 ± 1 | 337 ± 2 <br> 154 ± 1 | 355 ± 1 <br> 155 ± 1 | *** |
| **Dissolved oxygen (mg·L$^{-1}$)** | 7.9 ± 0.1 <br> 8.9 ± 0.1 | 7.8 ± 0.1 <br> 9.0 ± 0.1 | 7.7 ± 0.1 <br> 9.1 ± 0.1 | *** |

One-way ANOVAs for physicochemical parameters showed significant differences between Río Piedras and Río Sabana watersheds in temperature, pH, turbidity, conductivity, and dissolved oxygen (Table 3). Our results showed that stream water temperatures were higher in the Río Piedras watershed for all the sampling sites in comparison to the Río Sabana watershed. The lowest stream water temperature was observed in low urban reach in Río Sabana (21.3 ± 0.03 °C), while the highest water temperature was observed in

Río Piedras with high urban reach (23.9 ± 0.03 °C). In addition, significant differences in turbidity were observed among watersheds (Table 3). The low urban site in Río Sabana had the lowest concentrations of turbidity with a mean value of 0.1 ± 0.0004 NTU, while the highest mean value was observed in the Río Piedras watershed (0.2 ± 0.001 NTU). Water conductivity and pH values also varied significantly among watersheds (Table 3). One-way ANOVAs comparing pool canopy, pool depth, and substrate index among the Río Piedras and Río Sabana watersheds and between urban reach showed statically significant differences (Table 4).

**Table 4.** The mean (±SE) physical measurements in low urban (LU), medium urban (MU), and high urban (HU) reaches in urban (Río Piedras) and forested (Río Sabana) watersheds. NS—not significant at $p = 0.05$; *** $p < 0.001$.

| Physicochemical Variables | LU | MU | HU | ANOVA |
|---|---|---|---|---|
| | Urban Forested | Urban Forested | Urban Forested | |
| **Depth (m)** | 0.42 ± 0.004<br>0.36 ± 0.006 | 0.30 ± 0.003<br>0.37 ± 0.004 | 0.53 ± 0.05<br>0.40 ± 0.003 | *** |
| **Substrate index (SI)** | 1.52 ± 0.1<br>3.29 ± 0.13 | 1.20 ± 0.03<br>1.79 ± 0.069 | 1.19 ± 0.02<br>1.79 ± 0.05 | *** |
| **Pool canopy cover (%)** | 29.53 ± 0.99<br>39.69 ± 0.86 | 27.33 ± 0.99<br>34.01 ± 1.07 | 19.21 ± 0.67<br>22.41 ± 1.18 | *** |

*3.2. Freshwater Shrimp Sampling*

A total of 15,848 specimens of *Xiphocaris elongata* were collected in the watersheds. Higher numbers of Xiphocaris were observed in the forested watershed (N = 11,710) in contrast with the urban (N = 4138). For the forested stream, higher and lower numbers of shrimp were collected in June (N = 1123) in the LU reaches, and October (N = 94) in the HU reaches, respectively. At the urban watershed of Río Piedras, higher numbers of shrimp were reported in April (N = 316) at the LU reaches, and the minimum abundance in June (N = 24) at the HU reaches.

Significant differences were found in catch-per-unit efforts between the Río Piedras and Río Sabana watersheds (two-way ANOVA, $F_{(1,66)} = 42.8$; $p \leq 0.001$). Urban reaches in the Río Piedras and Río Sabana watersheds also demonstrated statistically significant differences in CPUE (two-way ANOVA, $F_{(2,66)} = 15.9$; $p \leq 0.001$) (Figure 2). In addition, statistically significant differences were found between the combination of urban reaches within Río Piedras and Río Sabana watersheds (two-way ANOVA, $F_{(2,66)} = 9.3$; $p \leq 0.01$) (Figure 2). Our results showed that high urban reaches in Río Piedras (4.4 ± 0.6 CPUE) and Río Sabana (7.4 ± 0.5 CPUE) had significantly different mean catch per unit effort. Similar results were found for medium and low urban reaches (Figure 2). Finally, the comparison of CPUE between HR and LR showed no significant difference for the Río Sabana watershed (Figure 3). However, in the Río Piedras watershed, significant differences were found between rainfall seasons (two-way ANOVA, $F_{(1,30)} = 50.1$; $p \leq 0.001$) (Figure 3).

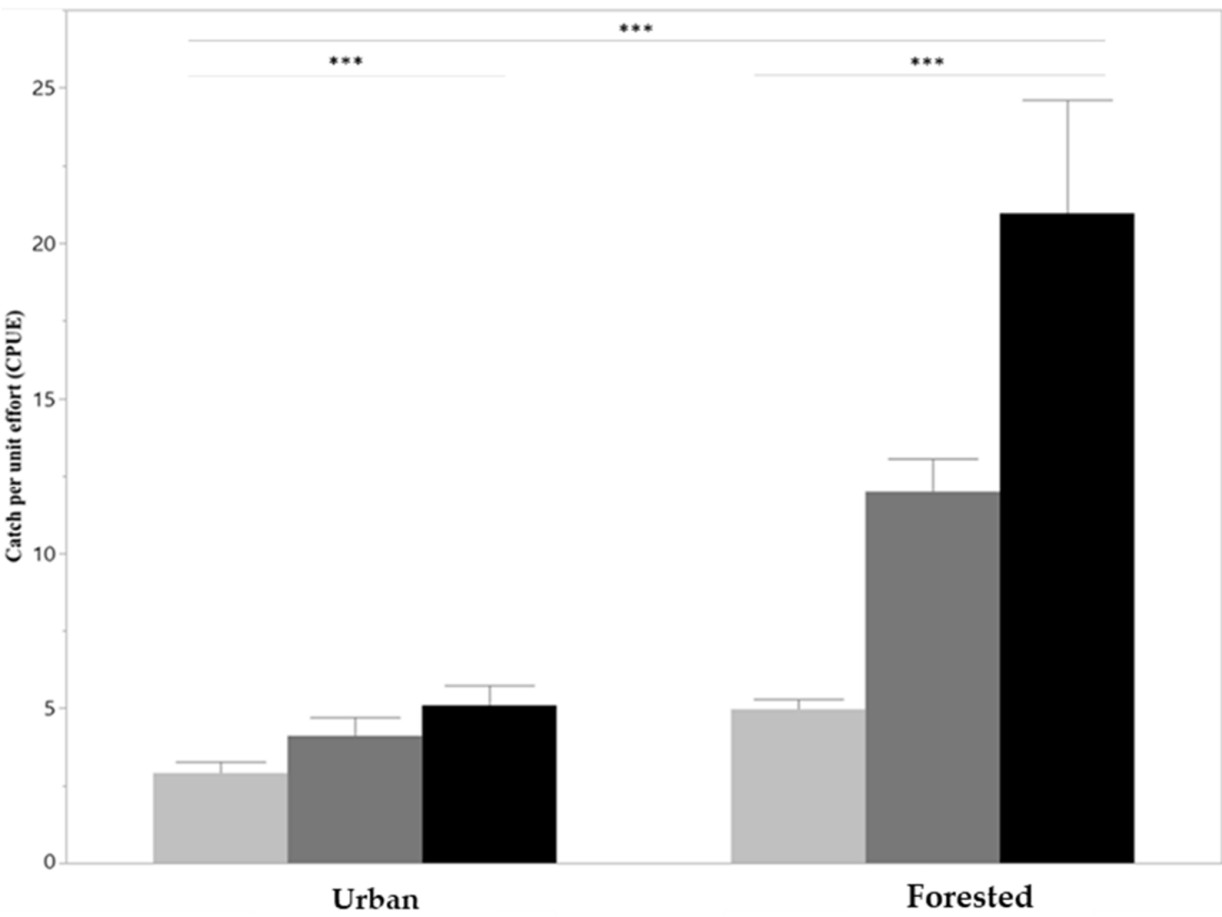

**Figure 2.** The mean (±SE) catch per unit effort (CPUE) for *Xiphocaris elongata* in urban (Río Piedras) and forested (Río Sabana) watersheds. Light grey bars represent HU (high urban), grey bars represent MU (medium urban), and black bars represent LU (low urban) reaches. The horizontal line over the bars represents the two-way ANOVA test differences *** *p*< 0.001.

### 3.3. Multivariable Analysis

Non-metric multidimensional scaling ordination of *Xiphocaris elongata* abundance in the Río Piedras and Río Sabana watersheds generated two major axes that explained most of the variation. In the Río Piedras watershed, axis 1 and 2 were highly related to turbidity (0.9) and pool depth (0.9), respectively (Figure 4a). Turbidity represented the environmental variable that explained most of the variation in *Xiphocaris elongata* abundance in the Río Piedras watershed. The environmental variable that best explained the variation on axis 2 was pool depth. Study sites with the deepest pools were observed in high (0.53 ± 0.05 m) and low (0.42 ± 0.04 m) urbanized areas. Other physicochemical variables such as pH and temperature were also important factors directly influencing *X. elongata* abundance in this watershed. In the Río Sabana watershed, axis 1 and 2 were highly related to temperature (0.9) and pool canopy cover (0.9), respectively (Figure 4b). Water temperature was the environmental variable that best explained the variation of abundance in this watershed. However, pH and turbidity variables can also influenced *X. elongata* abundance in the Río Sabana watershed.

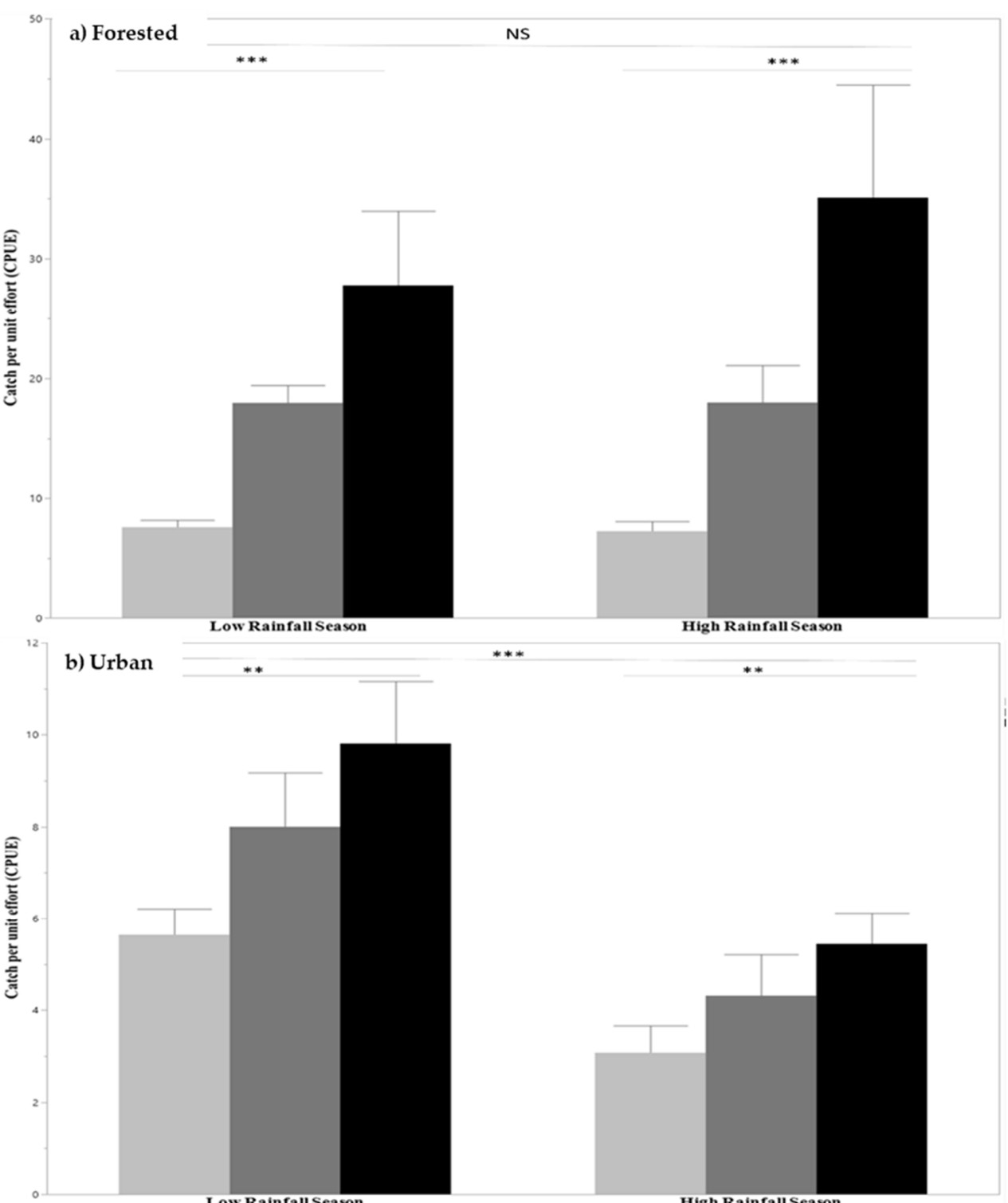

**Figure 3.** The mean (±SE) catch per unit effort (CPUE) for *Xiphocaris elongata* during LRF (low rainfall) and HRF (high rainfall) seasons. (**a**) Forested watershed (Rio Sabana) and (**b**) urban watershed (Río Piedras). Light grey bars represent HU (high urban), grey bars the MU (medium urban), and black bars the LU (low urban) reaches, respectively. The horizontal line over the bars represents the two-way ANOVA test differences NS—not significant; ** *p* < 0.01; *** *p* < 0.001.

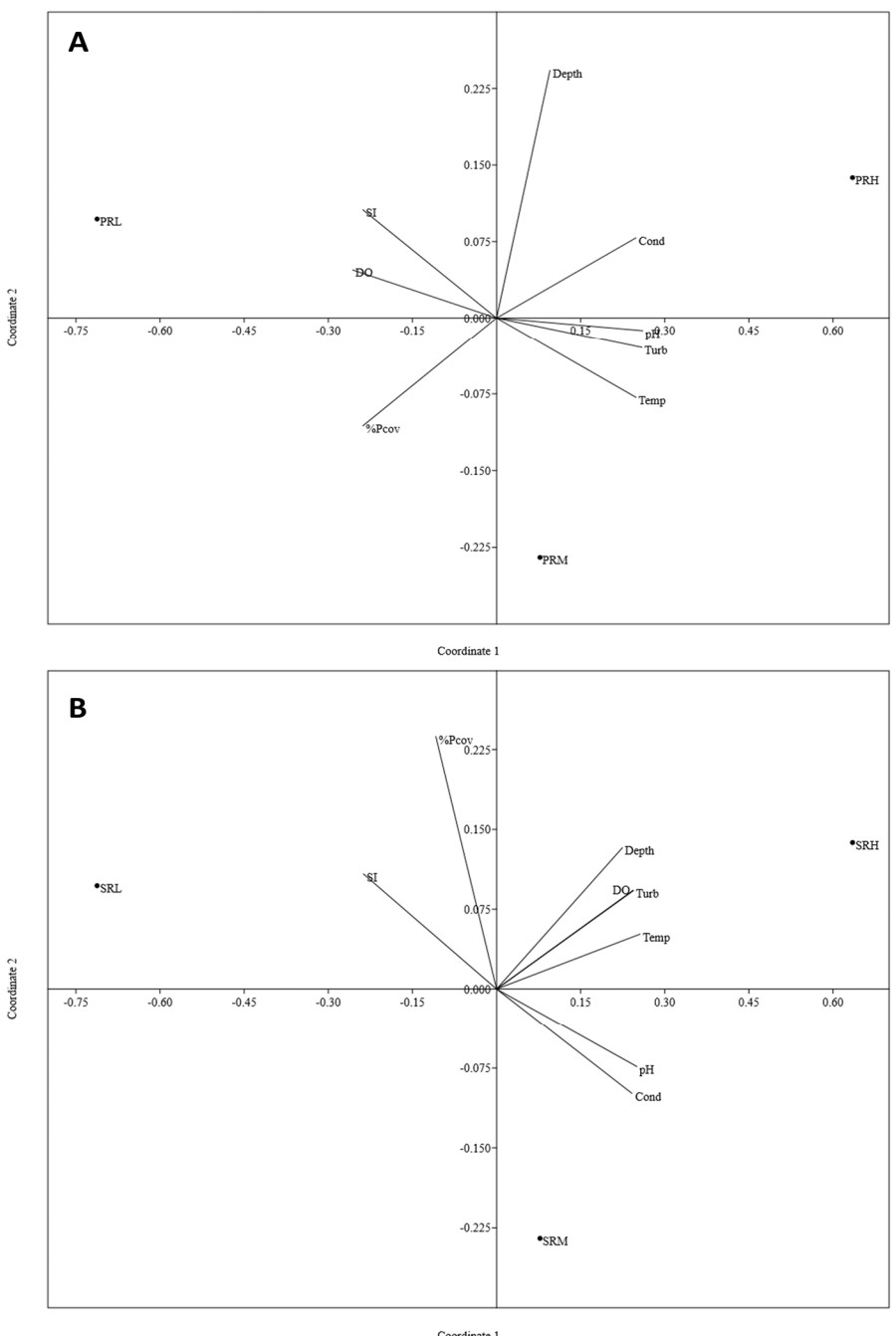

**Figure 4.** Non-metric multidimensional scaling (NMDS) results from the ordination of *Xiphocaris elongata* densities. Stream habitat variables were measured at the different sampling sites in (**A**) urban watershed (Río Piedras) (PRH-Piedras River high urban reach, PRM-Piedras River medium urban reach, and PRL-Piedras River low urban reach) and (**B**) forested watershed (Rio Sabana)(SRHSabana River high urban reach, SRM-Sabana River medium urban reach, and SRL-Sabana River low urban reach) were plotted as vectors. Vector length and direction reflect the strength and direction of the relationship between the stream habitat variables (Depth-pool depth, SI-substrate index, Pcov-%pool cover, DO-dissolved oxygen, Cond-conductivity, Temp-temperature, Turb-turbidity) and *Xiphocaris elongata* densities.

## 4. Discussion

Urban activities decrease the abundance of organisms living in freshwater environments [22,29,30]. However, it is not clear which environmental and anthropogenic variables

are most important for the observed decrease. This study found that urban reaches negatively affect the abundance of the freshwater shrimp *Xiphocaris elongata*. Consequently, these results demonstrated that this species is an excellent bioindicator to evaluate anthropogenic impacts on freshwater ecosystems. The observed result also confirms a previous study that found a decrease in species richness and abundance of decapod communities in highly urbanized watersheds in Puerto Rico [7]. We also found significant differences in catch-per-unit effort between HRF and LRF in the Río Piedras watershed. This difference could be related to extremely high stream discharge events that commonly occur in tropical urban streams. These events can scour stream channels and displace freshwater shrimp affecting their densities [30]. However, in forested streams, freshwater shrimp take advantage of high discharge events to transport their larvae downstream to the estuary to complete their life cycle (amphidromy) [14]. For the Río Sabana watershed, no significant differences were found between HRF and LRF seasons. Similar results have been found in forested watersheds where the abundance of decapod communities showed no variations between rainfall seasons [7,17].

The non-metric multidimensional scaling analysis revealed that the abundance of *X. elongata* in the Río Sabana watershed was best explained by temperature and pool canopy cover. Warmer temperatures were found in highly urbanized areas with lower canopy cover. Previous studies showed that higher water temperatures in urban stream environments are associated with reduced canopy cover [30–32]. The transformation of natural areas to impervious surface covers, such as roads and channels, can also facilitate heat transfer to freshwater environments due to the reduction in canopy cover [33]. The higher temperature in streams can increase decomposition rates and organism metabolism, reducing dissolved oxygen in the water. In the Río Piedras watershed, *X. elongata* abundance was best explained by turbidity and pool depth. In urban areas, higher turbidities were found in deeper pools. These results are consistent with symptoms associated with the urban stream syndrome [2]. A previous study in Puerto Rico found that increases in turbidity levels can have a negative effect on the migratory behavior of *X. elongata* [12].

## 5. Conclusions

It is evident that in tropical urban streams, human activities affect water quality, hydrology, geomorphology, and freshwater organisms. This study found that the highly urbanized reaches have less abundance of the shredder shrimp *Xiphocaris elongata* in comparison with reaches classified as medium or low urban in Río Piedras and Río Sabana watersheds. Our results demonstrate that physicochemical and geomorphological variables are important environmental factors that influence the abundance of *X. elongata* in Puerto Rican streams.

This study provides new knowledge about the anthropogenic impacts on the Caribbean region's freshwater ecosystems. This study demonstrated the benefit of using decapods as bioindicators to measure those impacts due to the abundance and dominance in tropical streams of shrimp rather than fish, as seen in temperate regions.

**Author Contributions:** Conceptualization, W.X.T.-P. and O.P.-R.; methodology, W.X.T.-P.; software, W.X.T.-P.; validation, W.X.T.-P. and O.P.-R.; formal analysis, W.X.T.-P. and O.P.-R.; investigation, W.X.T.-P. and O.P.-R.; resources, W.X.T.-P. and O.P.-R.; data curation, W.X.T.-P. and O.P.-R.; writing—original draft preparation, W.X.T.-P. and O.P.-R.; writing—review and editing, W.X.T.-P. and O.P.-R.; visualization, W.X.T.-P. and O.P.-R.; supervision, O.P.-R.; project administration, W.X.T.-P. and O.P.-R.; funding acquisition, W.X.T.-P. and O.P.-R. All authors have read and agreed to the published version of the manuscript.

**Funding:** This research was supported by a grant 1736019 from National Science Foundation (NSF) to the Neurobiology Institute, Medical Sciences Campus, University of Puerto Rico, San Juan. The authors declare no competing financial interests.

**Institutional Review Board Statement:** Not applicable.

**Informed Consent Statement:** Not applicable.

**Data Availability Statement:** The data presented in this study are available on request from the corresponding author. The data are not publicly available due to was part of a dissertation work of the student.

**Acknowledgments:** Fernando A. Villar-Fornés, and Marla Santos for their appreciated help in the search for materials for the research methodology. Thanks to the reviewers for their suggestions that increased the quality of the manuscript.

**Conflicts of Interest:** The authors declare no conflict of interest.

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
