# Peer review of "Population Status of the Tropical Freshwater Shrimp Xiphocaris elongata in Urban and Forest Streams in Puerto Rico"

_2673-9917, doi:10.3390/hydrobiology2010018_

Round 1

Reviewer 1 Report

Suggestions

1.     “a) Urban Watershed (Río Piedras), b) Forested Watershed (Río Sabana)”, are not indicated in Figure 1.

2.     Please pay attention to the format of Table 2 and Table3.

3.     There are no Icons in Figure 2 and Figure 3.

4.     Please describe “a “and “b” in the title of Figure 4.

5.     Please separate “Discussion” and “Conclusions” section.

6.     The authors need more discussions on the results, for example, the temperature is a natural factor of the stream, it should not be discussed to explain the different between the two stream.

Author Response

Thanks for your time, comments and suggestions. The last suggestion #6 about temperature was address in the discussion.

Reviewer 2 Report

Population status of the tropical freshwater shrimp Xiphocaris elongata in urban and forest streams in Puerto Rico

The objective of the present study was to characterize the population of X. elongata and to identify differences in the abundance of the species between urban and forest streams. As per the authors physicochemical and geomorphological variables are important environmental factors which influence the abundance of X. elongata in Puerto Rico streams.

The sampling design for calculation of shrimp is good and all the possible habitats were covered (riffles, runs, pools, and aquatic vegetation).

The range of variation of values of water quality parameters are very narrow, still, the differences are significant, which is very difficult to accept.

The population characteristic was one of the objectives, however, the result and the discussion are silent on that.

Author Response

Thank your for your comments, suggestions and time. We accepted all and did the changes in the manuscript.

Reviewer 3 Report

Review

Paper title: Population status of the tropical freshwater shrimp Xiphocaris elongata in urban and forest streams in Puerto Rico

The authors conducted a field study to reveal the abundance and CPUE of the freshwater shrimp Xiphocaris elongata at different locations in Puerto Rico. They found significant differences in these parameters between urbanized and forested areas and related these differences to environmental factors. They concluded that this species is a good ecological indicator for tropical streams.

All these reasons explain the relevance of the paper by Wesley X. Torres-Pérez and Omar Pérez-Reyes submitted to "Hydrobiology".

General scores.

The data presented by the authors are original and significant. The study is correctly designed and the authors used appropriate sampling methods. In general, statistical analyses are performed with good technical standards. The authors conducted careful work that may attract the attention of a wide range of specialists focused on biological indication of freshwater ecosystems.

Recommendations.

The authors should include coordinates (as a gird to the map or add in the text to the corresponding sites).

L 147. This source "Chace and Hobbs (1969)" is missing in the reference list.

The authors used ANOVA to test the data for differences. This parametric approach requires normal data distribution and data heterogeneity. Thus, the authors should test the data for normality and heteroscedasticity and transform the data if required or they should use a non-parametric approach.

The authors stated that they used NMDS but they presented an ordination plot that corresponds to PCA of PCoA (there are no stress values by factor loadings are presented). The authors should check and clarify the methodology.

Specific remarks.

L 68. Consider replacing “(Río Piedras),” with “(Río Piedras)”

L 130. Consider replacing “manufacturer instructions” with “manufacturer's instructions”

L 148. Consider replacing “Pérez-Reyes et al. 2013” with “Pérez-Reyes et al. [11]”

L 151. Consider replacing “length among” with “length of”

L 229. Consider replacing “a higher number” with “higher numbers”

L 242. Consider replacing “watershed” with “watersheds”

L 243. Consider replacing “statistically” with “statistically significant”

L 299. Consider replacing “elongata..” with “elongata.”

L 329. Consider replacing “watershed” with “watersheds”

Author Response

Thanks for your recommendations, we accepted all and include them in the manuscript.

Round 2

Reviewer 2 Report

no comments 

Reviewer 3 Report

The authors have revised the paper according to my comments.